# The Dual Strategy for Textile and Fashion Production Using Clothing Waste

**Hyewon Lee**

Department of Clothing & Textiles, The Catholic University of Korea, Bucheon-Si 14662, Republic of Korea; hyewonlee@catholic.ac.kr; Tel.: +82-2164-4322

**Abstract:** This study aims to utilize a dual physical and digital strategy for the completion of a process that achieves two goals: the treatment of large amounts of clothing waste and the development of materials and products. This study expands the author's previous research on the feasibility of using clothing waste as a textile material and the development of weaving methods. The processes of material analysis, design, material development, and product production for clothing waste were connected by the dual strategy. The project was conducted by three groups of designers for ten months and evaluated by ten experts. A total of eighteen digital products were developed, including three physical products and one digital twin. Digital and physical models were dressed and subjected to objective and in-depth evaluations by experts. The experts determined that the match rate between the physical products and digital twins was over 90% and that each process step was conducted appropriately. The process was also deemed applicable for 50% of the industrial sector and 80% of the education sector. Therefore, this study connected the quantitative disposal of garment waste to the qualitative design and production of new material, introducing a new process strategy to maintain sustainability in the fashion industry.

**Keywords:** dual strategy; clothing waste; sustainable process; circular fashion

## 1. Introduction

### 1.1. Background

Clothing waste is a major contributor to environmental pollution and has been considered unavoidable in the fashion industry. Existing disposal methods for clothing waste, such as incineration and landfilling, have problems such as cost burden for waste disposal [1,2], treatment of pollutants generated during the disposal process [3–7], and lack of landfill sites [8–10]; therefore, the development of sustainable clothing waste treatment and recycling methods for environmental circulation is a great social and industrial need.

Thus far, producers and consumers have practiced conventional recycling methods. Producers use methods to reduce the generation of clothing waste at the sources. For example, in the fiber and fabric production stage, the post-consumer disposal phase of the product is considered in advance, and measures are developed to reduce the pollution caused by incineration and landfill disposal, such as material changes and improvements [11,12]. At the design level, methods to achieve minimal or zero percent fabric waste in product manufacturing have been devised, including prototype development using 3D virtual digital technologies [13–17]. At the product sales and consumption stage, sellers have developed and implemented strategies to sell their products using ethical business practices [11,12,18–21], whereas consumers have sought to focus their attention on recycled fashion products and to improve their indiscriminate consumption patterns [11,12,22–24]. Compared with producers, the number of ways in which consumers contribute to helping the environment is limited. Consumers, the end-users of fashion products, have little knowledge of clothing waste recycling and lack specialized skills for it. Therefore, consumers mainly reduce their overconsumption of new products, reusing them instead of

discarding, or trading used items [25]. As fashion trends change rapidly with the growing demand for recycling clothing waste, the amount of clothing waste generated by end-users eventually increases despite the technological efforts of producers to reduce them [25–27].

Reducing waste through quantitative and qualitative recycling for producers and consumers requires more actionable methodologies because at the end of the product life cycle (PLC), during the disposal phase, only a small percentage of clothing products ($\leq$20%) that have outlived their usefulness are considered recyclable [25,26]. Even then, only about half of them are reused for recycling and second-hand fashion, while the rest end up in landfills [25]. The ultimate challenge in the fashion textile industry is the creation of a flexible and appropriate link between the last stage of the PLC, in which waste is generated, and the first, the material stage. Therefore, a methodology that can reduce the amount of waste while simultaneously reducing the purchase of raw materials in the first stage of product production must be devised, that is, to actively utilize waste as a material for product production and to pass it through the PLC once more [25–31].

The European Union (EU) has also been calling for new methods to reduce clothing waste other than incineration and landfills. The EU strategy report released in March 2022 [32] officially announced that the future challenge was to digitally realize the sustainability of the textile ecosystem. In the report, the "Sustainable Circular Textiles Strategy" outlined the link between fast fashion and fossil fuels and the urgency of introducing alternatives to change the textile ecosystem. In particular, the concern for "chemical recycling", which has been attempted by the textile materials industry, possibly causing environmental pollution, was raised. Therefore, proposing a twin transition or double transition that prioritizes fiber-to-fiber recycling and achieves sustainable circular textiles together with digital is necessary.

A dual strategy or transition refers to an approach or action plan designed to achieve two goals simultaneously or to address one goal using two methods [33–39]. The fashion industry can use two possible twin strategies to address the problem of clothing waste disposal. The first can be described as the achievement of two goals: clothing waste reduction and textile fashion production. The second can be described as the simultaneous use of physical and digital methods to achieve a single goal: environmental circularity [37–39]. In the fashion industry, twinning strategies targeting clothing waste have been limited. Although physical strategies have been developed to clean and recycle clothing waste into materials for new products [4–7,9,12,14,20], few have utilized digital methods in this process. In addition, many fashion companies use digital prototyping strategies to reduce waste in the prototyping process [38,39], and physical product prototyping strategies are becoming increasingly distant from green strategies.

The beginning of fashion production is often a digital prototype, and the end is a physical product that has been used and lost value. To connect the end and beginning stages flexibly, the product strategy for each stage's deliverable should be interconnected and complementary. To date, the digital prototyping and design stage has been the furthest from the disposal of physical products, with challenging and experimental prototyping based on free-form ideas using infinitely new raw materials in a digital environment. The technical advantages of a digital strategy do not translate into the later stages of physical production and inventory handling, and even less so for product disposal. As stated in the EU report [32], the need to develop strategies to transition the end of the fashion stream from simple incineration and landfilling to fiber-to-fiber recycling and digital textiles from physical waste is urgent. The final stage of the product, that is, the quantitative treatment of physically generated waste, should be linked to the first stage of the product, the qualitative design process of product development. In other words, the direction of materializing waste and using it as a raw material for products should be explored, and a twin strategy should be implemented to increase the viability of the strategy. In the future, the fashion industry will need to implement the first twin strategy, which connects the garment disposal phase with the garment design phase, and the second twin strategy, which involves the simultaneous implementation of physical and digital methods to connect the two phases.

Achieving the twin strategy is not only an alternative to the traditional disposal of clothing waste in landfills and incineration, but also a way to complete the sustainable cycle of the fashion industry, which has been accelerated by digitalization.

### 1.2. Research Objectives and Research Process

The objective of this study was to link the garment disposal stage to the garment material development stage for a sustainable fashion stream. This study is a follow-up to two previous studies [40,41] that were conducted to develop a plan to utilize physically generated clothing waste as apparel material, and utilize low-grade waste and improve its quality to regenerate its use value as a product. In the first preliminary study, the authors collected a variety of clothing waste and investigated their utility as weaving material through simple shape modification, and simultaneously designed various weaving methods [40]. In the later stage, to increase the utilization of clothing waste materials and the feasibility of recycling methods for the public and overcome the limitations of the previous study, we refined the weaving method, simplified the material preparation, produced woven fabric samples, and produced virtual digital textile prototype products [41]. The purpose of this research was to simultaneously achieve the objectives of reducing clothing waste and conceiving materials for the production of new products, conduct a dual physical and digital strategy to produce materials and products from clothing waste, and perform the strategy step-by-step to complete the entire process. The research problem was divided into four phases: material analysis, design, material development, and product manufacturing, with two dual strategies for each phase.

1. Material analysis stage: Analyze the conditions of clothing waste and identify the possibility of physical and digital productization.
2. Design stage: Based on material analysis, predict possible product types and plan the design for physical and digital products.
3. Material development stage: Produce physical and digital textiles and materialize physical and digital products.
4. Product manufacturing stage: Complete the production of physical and digital products and conduct a final review of the results and processes of the dual strategy.

The schematic of the process executed in this study is shown in Figure 1, and the study content for each phase is as follows: First, the material analysis stage is a preparatory step that connects the concurrent goals of waste disposal and product development strategy, and is an important first step towards full circularity, which is the goal of this study. At this stage, the researcher collects clothing products that are about to be discarded, and determines whether they can be recycled as materials for physical or digital products. Second, in the design stage, products are designed based on the data analyzed in the previous phase, with product types divided into digital and physical. The third stage of material development involves weaving cloth from clothing waste for direct use as a component in physical and digital products. This stage includes swatches and textile production through weaving. Physical and digital products are manufactured during the product manufacturing stage and the products are evaluated. The products and processes are evaluated in depth by a panel of expert judges recruited for the review.

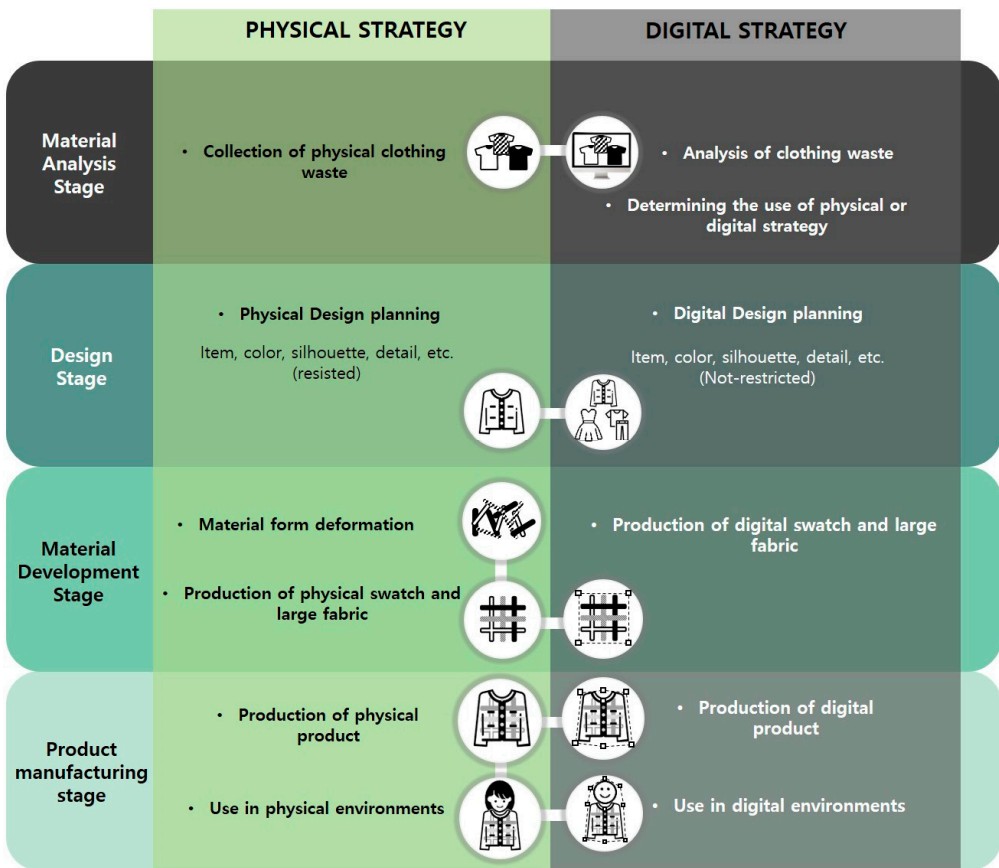

**Figure 1.** Conceptual diagram of the process showing the implementation of the dual strategy in the material analysis, design, material development, and product manufacturing stages.

## 2. Literature Review

### 2.1. PLC for Using Clothing Waste as a Material

Clothing waste is generated randomly in many locations and in many different types, and is more post-consumer than pre-consumer [2,18,22–24,28], an inevitable consequence of changing consumption patterns due to rapidly changing fashion trends. Although experts estimate the recycling rate of fashion textiles to be over 95%, in the real world, a significant portion of apparel waste ends up in landfills [29,30]. The disposal of significant environmental pollutants such as toxic gases from landfills and incineration is a major issue in the garment disposal process, and various efforts are being made by the industry to reduce the amount of clothing waste that is inevitably generated as a fashion product.

The alternatives chosen by industries are primarily to reduce clothing waste before consumer use [29,31,42–44]. This strategy attempts to improve the environment by initially improving the materials and processes that contribute to clothing waste. In contrast, the alternatives chosen by consumers are mainly to reduce the consumption of new goods or to sell and buy used clothing [24,31]. Consumers sell clothing waste with good design and quality conditions to other consumers as "second-hand" products [24,28,31,43]. Used clothing has a shorter product life than new clothing, which means that it will end up in landfills or dumps sooner than new items. Consumers discard garments for various reasons, including soiling and staining, body size changes, deterioration, functionality limitations, poor taste, and outdated style. All these reasons inevitably occur over time and become more difficult to overcome, eventually leading consumers to opt for the "easy" disposal methods of incineration and landfilling.

One way to drastically reduce the amount of clothing waste generated in the post-consumer stage is through the materialization of clothing waste. Connecting the clothing waste stage with the garment material development stage can extend the life cycle of a



product by connecting the final and initial stages of the fashion product production stream. Previous studies have shown that the materialization of clothing waste has been continuously attempted [29,43–45]; however, solving problems such as adjusting the product price, which is higher because of the cost of expert input, and changing the stereotype that the product must be of low grade because it uses waste materials has been difficult. The materialization of clothing waste, which should be practiced only by professionals, is expected to slow down the achievement of circularity in the fashion textile industry eventually, leading to the conclusion that clothing waste recycling should be practiced not only by professionals but also by consumers, who are the main generators of final clothing waste [45].

The final goal of this study was to develop a dual strategy to achieve the goals of waste treatment and materialization using both digital and physical methods. Figure 2a shows the life cycle assessment (LCA) of a typical fashion product [21,31,46] surrounding the issue of clothing waste disposal, and Figure 2b shows the product LCA that is expected to change if the "dual strategy" pursued in this study is implemented. In a typical LCA (Figure 2a), the stages relevant to clothing waste disposal are primarily related to design, product concept and design, raw material sourcing, and manufacturing. The product concept and design stage is the first step in the PLC, where products are designed based on the needs of designers or consumers; the demand for "new" and "trendy" designs often leads to digital and experimental design attempts. The prototyping process is followed by the purchase of raw materials to create an optimal product design from new materials. Figure 2b shows the LCA of a fashion product that utilizes clothing waste to implement a twin strategy designed to reflect the goals of this research. For a sustainable fashion industry and the circularity of resources, recycling should be the first step in the fashion stream and not the last. In this new LCA, clothing waste generated at the end of the physical strategy was used as a material in the product, affecting both its design and production. Clothing waste, which is not constant in quantity and type, can be a positive material for product design, and the two strategies can be implemented simultaneously or selectively, depending on the clothing waste material. The stages of design, materials, and products have a two-way interaction, and as the strategy is implemented according to the clothing waste situation, clothing waste materials are used. The main advantage of this process is that, unlike traditional LCA, no raw material purchases are made, thus saving time on material searching and expenses. The purpose of this process is to create new materials from clothing waste without the material purchase stage, and to use them in products.

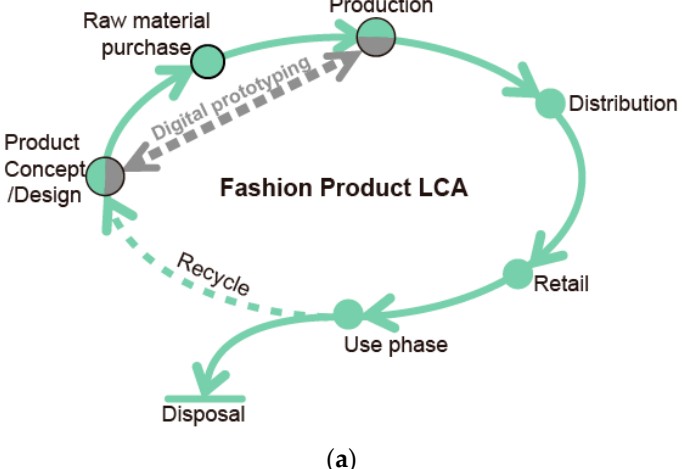

(**a**)

**Figure 2.** *Cont.*

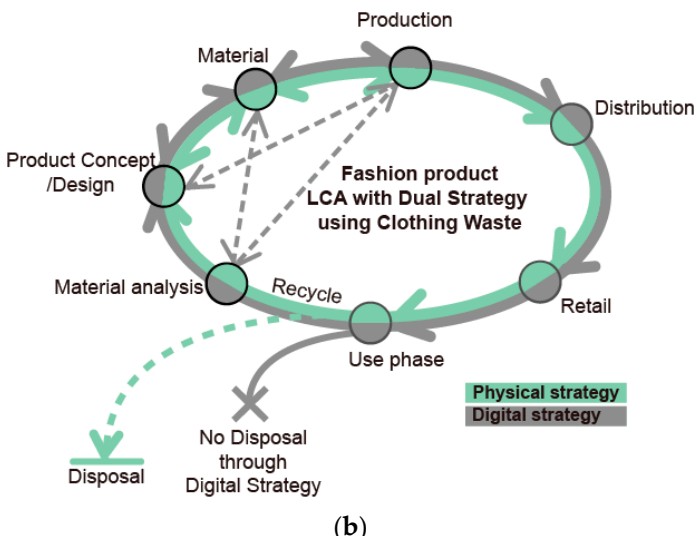

(**b**)

**Figure 2.** Life cycle assessment (LCA) of a (**a**) typical fashion product and (**b**) fashion product with dual strategy using clothing waste.

*2.2. The Steps of Previous Research and the Significance of This Research*

In previous research, the author searched for ways to make recycling clothing waste easier and more aesthetically pleasing at the consumer level. The author's first study considered the development of a method to explore the possibilities of converting clothing waste into textiles through weaving, and the second concerned the creation of textiles from clothing waste that are both aesthetically pleasing and practical. In the first study [40], the author found through preliminary research that most end-users who generate clothing waste are unaware of how to recycle it but would like to know how to do so. The author also found that people believe recycling clothing waste should be easy and convenient and that the recycling process should be enjoyable. Based on these opinions, the author developed weaving methods that were easy to complete, fun to learn, and easy to practice and produced weaving samples using various clothing waste materials [41]. The limitations of the first research finding are as follows: First, because clothing waste comes in a variety of shapes and designs and is generated randomly rather than consistently, experts have stated that material preparation must be simple for waste to be consistently and easily utilized as a product material in large quantities. This method requires turning a variety of clothing waste into the same shape for easier weaving. Finally, the practicality and applicability of textile samples made from clothing waste must be improved.

In the second study [41], the authors focused on overcoming the limitations of the previous research. Thus, the authors succeeded in modifying and improving the weaving method, unifying the material form through simple cutting rather than through the difficult dismantling of the product, overcoming the sample size of physical textiles, and expanding its practicality through virtual 3D digital textualization. In this study, textile results were objectively evaluated by dividing them according to novelty and appropriateness, and a focus group interview was conducted with experts majoring in 3D virtual digital clothing design to qualitatively evaluate the entire process and results.

This study represented the third step in the production and completion of the entire process. Physical and digital fabrics were manufactured to complete the entire process, based on a clothing waste-based weaving method developed in the previous study. In this study, physical and digital textiles are produced simultaneously or selectively from physically generated waste materials and they serve as the basis for which physical and digital products will be produced. The use of digital/digital strategies at each stage of production is determined by the material situation of the clothing waste, and a digital product is created simultaneously as a twin for the physical product. In addition, the physical product, which is a single product, is expanded through various design variants

along with its digital twin. What distinguishes this research from other studies that use 3D virtual digital technology to create fashion products [33–39,47–51] is that clothing waste material is used to solve the dual problem of waste disposal and product development using both physical and digital strategies simultaneously, which is the significance of this research. This study aims to increase the efficiency of clothing waste disposal and to achieve full circularity by connecting the steps from clothing disposal to material development to product production and providing an integrated discussion of how the simultaneous implementation of physical and digital strategies can be a positive and forward-looking alternative to clothing waste disposal.

## 3. Methodology

### 3.1. Process Subjects

For this study, subjects must experience the entire process of collecting, designing, materializing, and producing clothing waste. Therefore, the project was implemented by a group of designers (five females and five males) who majored in fashion design. Students majoring in design were chosen as subjects for this study because they have knowledge of basic fashion design, interest in sustainability, and the ability to be creative and artistic. The students were informed of the research process in advance, and the space and materials for the product were provided by the author. The entire process lasted for 10 months, from September 2022 to June 2023.

### 3.2. Procedure and Methods

3.2.1. Material Analysis Stage

Clothing waste, with no restrictions on type, were voluntarily collected from subjects one week prior to the project. At this stage, the amount, type, color, size, and functional conditions of the collected clothing waste were checked, and the size of the fabric that could be woven for the material development stage was calculated. The calculation of the woven fabric size was based on the horizontal length (top: chest width, bottom: waist circumference) and vertical length (top: body length from the highest point of the shoulder, bottom: length including the belt) of the clothing waste. In the case of tops among clothing waste, it was calculated based on the body part, excluding the sleeves, and the calculation result of the fabric size of the body part was deemed as the minimum fabric size that could be woven.

The calculated fabric width was converted to "cm", because the total length of fabric tape can be cut into 1 cm wide strips for weaving. The reason for the unit conversion was to calculate the area of the fabric made from clothing waste compared with the area of a yard of commercially available 45-inch-wide basting fabric. This is necessary because the area of the fabric that can be woven determines the use of physical and digital strategies and types of clothing products that can be created, which can help with design planning. When weaving with randomly collected clothing waste, if a large amount of the same type of waste material was used, a large physical fabric could be woven, which is suitable for physical productization. If the waste material was different and the amount was small, only small-sized physical textile swatches could be made and digitized or made into small accessories.

3.2.2. Design Stage

In the design stage, the design of a digital or physical product is specifically planned based on the data analyzed in the previous stage. For a physical product, the design must accurately reflect clothing waste, colors, patterns, and textures. However, a digital product may have the same design as a physical product, and digital technology allows for more freedom in design variations because colors, materials, patterns, and more can be changed and added. The design tool does not suggest a physical/digital method and allows both hand-drawn and digital drawings. Three groups of designers planned their own designs according to digital and physical strategies, considering the color, texture, and weave of

the fabrics that could be made from the selected clothing waste materials. The weave was based on three of the highest-scoring methods from the seven methods developed in a previous study [41]: plain, weft irregular, and matt irregular.

### 3.2.3. Material Development Stage

Based on the previously planned design, this stage involves the direct creation of fabrics using clothing waste as the material. Specifically, a physical fabric swatch was first created, and then a large physical fabric and a digital fabric were created based on it. To create the physical fabric swatches, weaving frames (size: small; material: wood; width: 165 mm; height: 210 mm; interval between wraps: 8 mm) (Figure 3a) were used, as in previous studies [40,41]. The swatch was 13 cm wide and 13 cm tall and remained within the weaving frame. The reason for this square is that it can be registered as a unit and repeated in digital textile production. The designers were informed that the clothing waste could be cut into tape, as shown in the examples of the cutting directions in Figure 4, and they were free to prepare their materials. The fabric to be applied to the digital product was digitized after making the physical textile swatch, and the fabric to be applied to the physical product was made into digital and physical textiles simultaneously after making the physical textile swatch. Two types of weaving frames (size: medium; width: 500 mm; height: 700 mm; Figure 3b; and size: large; width: 310 mm; height: 430 mm; intervals between wraps: 3 mm; Figure 3c) were used for large-sized physical textiles. The Adobe Substance 3D Sampler program was used to create the digital textiles (Figure 5).

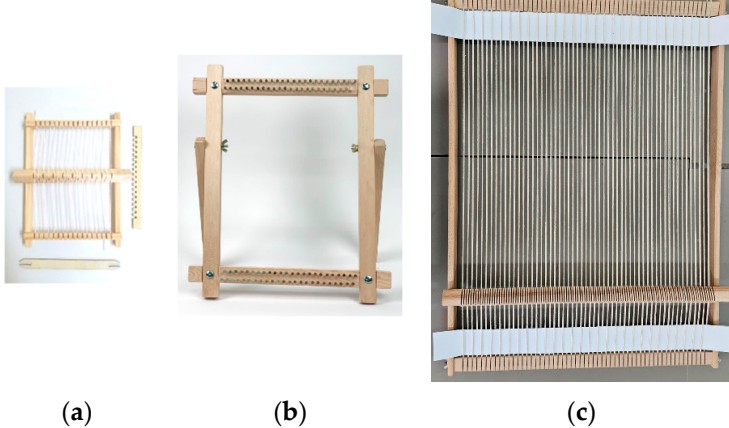

(**a**)　　　　　　　　　　(**b**)　　　　　　　　　　(**c**)

**Figure 3.** Weaving frames: (**a**) small-size frame for the development of physical fabric swatches and (**b**) medium-size and (**c**) large-size frames for the development of physical fabrics.

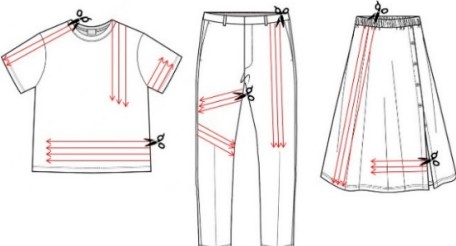

**Figure 4.** Cutting directions sample for clothing waste.

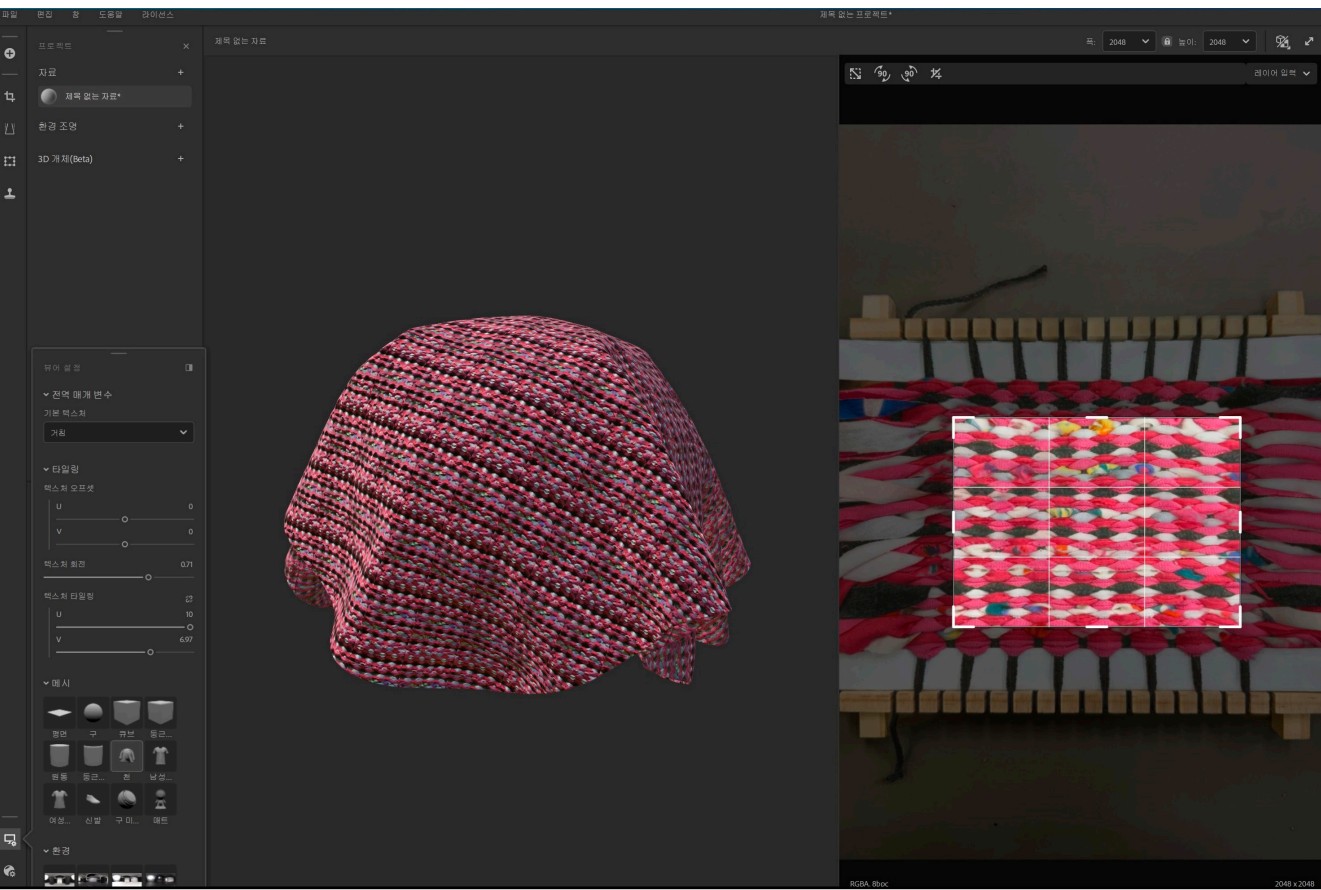

**Figure 5.** Screenshot of the Adobe Substance 3D Sampler program used to create digital textiles.

### 3.2.4. Product Manufacturing Stage

The products were manufactured using physical and digital fabrics based on previously developed textile swatches. All three groups of designers created five digital products in addition to at least one physical product and one digital twin product. The program used to create the digital product was CLO3D. Pattern drawings for the physical products were created using computer-aided design digital patterns within CLO3D. After production was complete, the product was placed on a person. The wearer's size was that of the recruited volunteers whose physical measurements were taken and applied to the digital avatar's size.

### 3.2.5. Evaluation

The process and products completed up to the product manufacturing stage were evaluated objectively and reviewed in-depth by ten experts. The ten experts comprised five professors and five designers (Table 1). The experts have at least 10 years of specialized knowledge and experience in fashion design, materials, and products, and were active in both domestic and international research and industry. An objective design evaluation compared the degree to which the design elements of the physical product and digital twin matched. This evaluation focused on the material and product stages and evaluated the matching of the following design elements: color, texture, and woven pattern of the material, structural form and details of the product, and outfit silhouette parts. The judges were asked to rate the congruence of the products created by the two strategies on a 9-point Likert scale ranging from 0% to 100%, with a 12.5% interval. The in-depth evaluation used expert focus group interviews. The experts discussed their evaluation of the execution of each step, appropriateness of the dual strategy for the overall process, and utility of the

process. Table 2 lists the evaluation subjects, factors, and methods for the objective design and in-depth evaluations.

**Table 1.** List of experts who participated in the objective design and in-depth process evaluations.

| Expert | Occupation | Education | Major | Work Experience | Nationality |
|---|---|---|---|---|---|
| A | Professor (academy) | Doctoral degree | Fashion design | 10 years | South Korea |
| B | Professor (academy) | Doctoral degree | Fashion design | 12 years | South Korea |
| C | Professor (academy) | Doctoral degree | Fashion design | 12 years | South Korea |
| D | Professor (academy) | Doctoral degree | Clothing materials | 20 years | United States |
| E | Professor (academy) | Doctoral degree | Clothing materials | 23 years | United States |
| F | Designer (industry) | Bachelor's degree | Woman's wear | 11 years | South Korea |
| G | Designer (industry) | Bachelor's degree | Unisex wear | 11 years | South Korea |
| H | Designer (industry) | Bachelor's degree | 3D virtual garment | 7 years | China |
| I | Designer (industry) | Master's degree | Sports wear | 15 years | South Korea |
| J | Designer (industry) | Master's degree | 3D virtual garment | 9 years | South Korea |

**Table 2.** List of the evaluation subject, factors, and methods for objective design and in-depth evaluations.

| | Evaluation Subject | Evaluation Factor | Evaluation Method |
|---|---|---|---|
| Objective design evaluation | Physical product and digital twin product | • Color of the material<br>• Texture of the material<br>• Woven pattern of the material<br>• Structural form of the product<br>• Detail of the product<br>• Outfit silhouette | 9-point Likert scale (0 point: 0%; 9-point 100%; interval: 12.5%) |
| In-depth evaluation | The process | • Each stage: · Material analysis stage · Design stage · Material development stage · Product manufacturing stage<br>• Overall process · Appropriateness of dual strategy · Utilization of the process | Focus group interview |

## 4. Results

### 4.1. Material Analysis Stage

One week after collection, 73 pieces of various clothing waste, excluding the most contaminated, were determined to be potentially usable as materials. Clothing waste was organized according to type, fabric composition, marked size, color, and photographs (Table 3). They were divided into t-shirts and knitwear for tops, pants, skirts, and dresses for bottoms, and accessories, with t-shirts being the most common. The clothing waste contained a wide range of fiber mixes, with cotton being the most common, and varied in color and printed pattern. The clothing waste materials were all woven fabrics, except for four knitted garments. The collected clothing was discarded for the following reasons: it was out of fashion, unwearable, and non-functional.

**Table 3.** Organization of clothing waste by type, material composition, labeling, color, and illustration.

| Item (number) | T-Shirts (33) | Knitwear (4) | Pants (24) | Skirt/Dress (8) | Accessory (4) |
|---|---|---|---|---|---|
| Type | - Sleeveless (2) <br> - Short sleeve (18) <br> - Long sleeve (13) | - Long sleeve (3) <br> - Short sleeve (1) | - Long pants (19) <br> - Short pants (5) | - Long skirt (4) <br> - Short skirt (2) <br> - Long dress (2) | Scarves (4) |
| Fabric composition | Cotton, polyester, polyurethane, acrylic, rayon, silk, and no info | Wool, acrylic | Cotton | Cotton | Cotton |
| Marked size | Small, medium, large, free | Small, medium, free | Small, medium, large | Small, medium, free | 50 × 50 (2) <br> 70 × 200 (1) <br> 90 × 90 (1) |
| Color | White (4), Yellow (2), Orange (1), Green (2), Pink (5), Purple (2), Brown (1), Black (5), etc., (Patterned 11) | White (1), Black (1), Beige (1), Pink (1) | Blue denim (12), Black (3), Pink (1), White (1), Yellow (1), Khaki (1), Gray (1), Patterned (4) | Blue denim (1), Yellow (1), Green (1), Patterned (5) | Pink (1), Patterned (3) |
| Sample Photos |  |  |  |  |  |

The weavable size was estimated based on the material analysis of the collected clothing waste. The amount of fabric that could be used for weaving had to be calculated individually for each piece of clothing waste collected because they all had different silhouettes, lengths, types, and amounts of products. The minimum area of fabric that could be woven from clothing waste was calculated, excluding knitted garments. The calculated fabric width was converted into the total length of the fabric tape that could be cut into 1 cm wide strips for weaving. Assuming that the area of the 45-inch-wide and 1-yard-long fabric was 100%, the percentage of fabric that could be woven from the clothing waste was calculated. Based on the calculated results, the products expected to be manufactured were derived, as listed in Table 4. The results of the analysis in this stage were used as a basis for the next design stage, simultaneously considering which clothing waste material to choose for weaving and what kind of product to design. In the case of short clothing waste, crop tops and shorts could be produced, whereas various garments, including outers, could be produced from long garments.

*4.2. Design Stage*

At this stage, the products were specifically designed based on the textile sizes estimated in the previous material analysis stage for weaving to be performed in the later material development stage. Based on the color, texture, pattern, and weave availability of the selected materials identified in the previous stage, designers planned the product according to digital or physical strategies. The design tools used by all three groups were the digital programs Procreate (iPad) and Adobe Illustrator. Designers primarily used Procreate for rough sketches, with precise schematics redrawn in Adobe Illustrator. In the design stage, a physical product was designed to reflect the characteristics of the collected clothing waste, and various digital products, including a digital twin of the physical product, were designed together. Figure 6 shows the results of the design stage.

**Table 4.** Summary of results of calculated material conversion for weaving by type of clothing waste.

| | Top (T-Shirt, Knitwear) | | | Pants | | Skirt | | Dress | Accessory |
|---|---|---|---|---|---|---|---|---|---|
| | Sleeveless (2) | Short Sleeve (19) | Long Sleeve (16) | Short Pants (5) | Long Pants (19) | Short Skirt (2) | Long Skirt (4) | Knee Length (2) | Scarves (4) |
| Total length of the 1 cm wide tape format (cm) | 4383.7 | 5610.5 | 5881.0 | 3400.3 | 6408.6 | 2814.4 | 5269.1 | 9642.4 | 2500 ~14,000 |
| Average percentage of the area compared with one yard of fabric (45 inches wide) | 41.9% | 53.7% | 56.27% | 32.5% | 61.3% | 26.9% | 50.4% | 92.3% | 23.9 ~134.0% |
| Products expect-able to manufacture | Bottom (short), top(crop), vest(half) | Outer (body), bottom (short), vest (more than half) | Outer (body and short sleeves), top, bottom, (above the knee length), vest | Bottom (less than short length), vest (less than half) | Outer (body), top, bottom (knee length), vest | Bottom (less than short length) | Outer (body), bottom (short), vest (more than half) | Outer, top, bottom, vest, etc. | - |

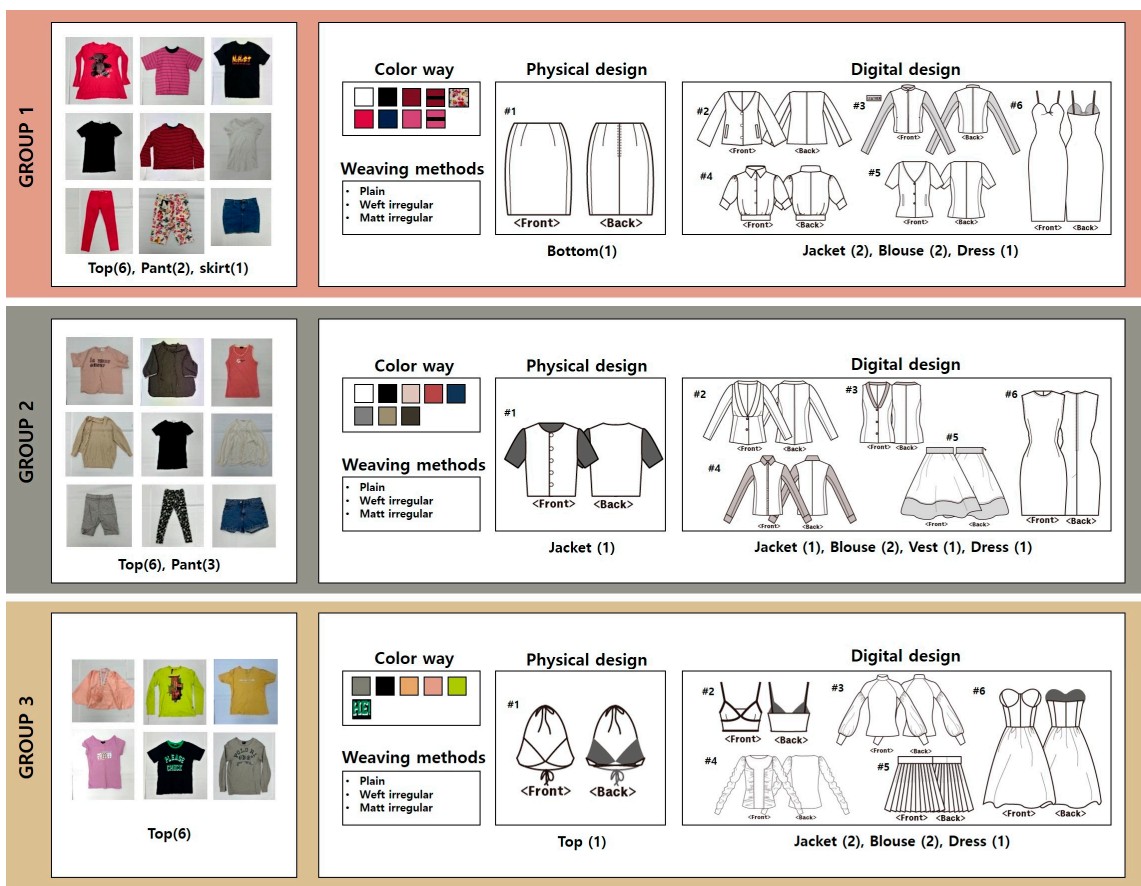

**Figure 6.** Results of the design stage: chosen clothing waste materials, color ways, weaving method planned, and technical drawings of physical and digital designs.

### 4.3. Material Development Stage

Based on the previously planned design, each group of designers cut the planned clothing waste to be used to create the physical fabric swatches into tape format and prepared them for weaving, as shown in Figure 7. Physical fabric swatches were created (Figure 8a) and digitally fabricated. In addition, swatches that were intended to be made into larger fabrics and converted into physical products were rewoven on a large weaving frame (Figure 8b). Each group created physical fabric swatches, digital fabrics, and physical fabrics from selected clothing waste. A total of 13 physical swatches were created, all of which were used as swatches for the digital fabric to create a larger fabric. The three digital fabrics were made into physical fabrics. Figure 9 shows the results at this stage.

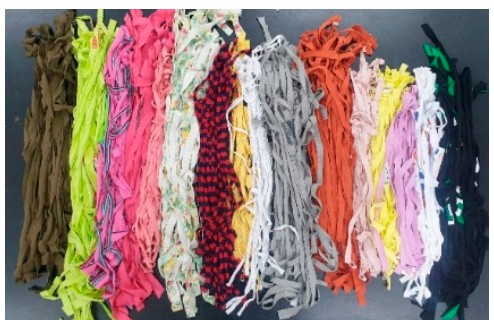

**Figure 7.** Clothing waste cut into 1 cm wide tape format.

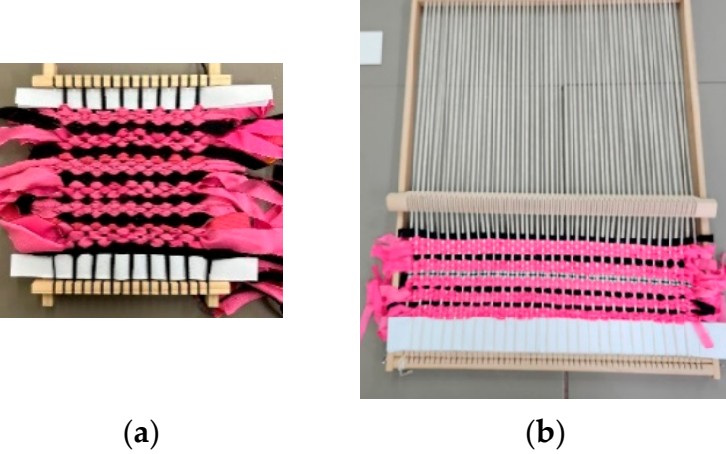

(**a**)                                    (**b**)

**Figure 8.** (**a**) A fabric swatch woven on a small weaving frame (**b**) and the physical large fabric woven on a large weaving frame.

### 4.4. Product Manufacturing Stage

From the fabrics created in the previous stage, eighteen digital products and three physical products were created using the physical fabrics and their digital twins. Three digital products were twins of the physical products. The final digital items included four jackets, one bustier bra, one sleeveless top, five blouses, one vest, three dresses, and three skirts. Physical products included skirts, jackets, and bustier bra, all of which have digital twins. The completed products were worn by humans and avatars, and the worn state, were photographed from six angles (front, back, right, left, three-quarters left, and three-quarters right) and used for expert evaluation. Figure 10 shows the finished product for each group, and Figure 11 shows a comparison of photographs of the human and the avatar wearing it.

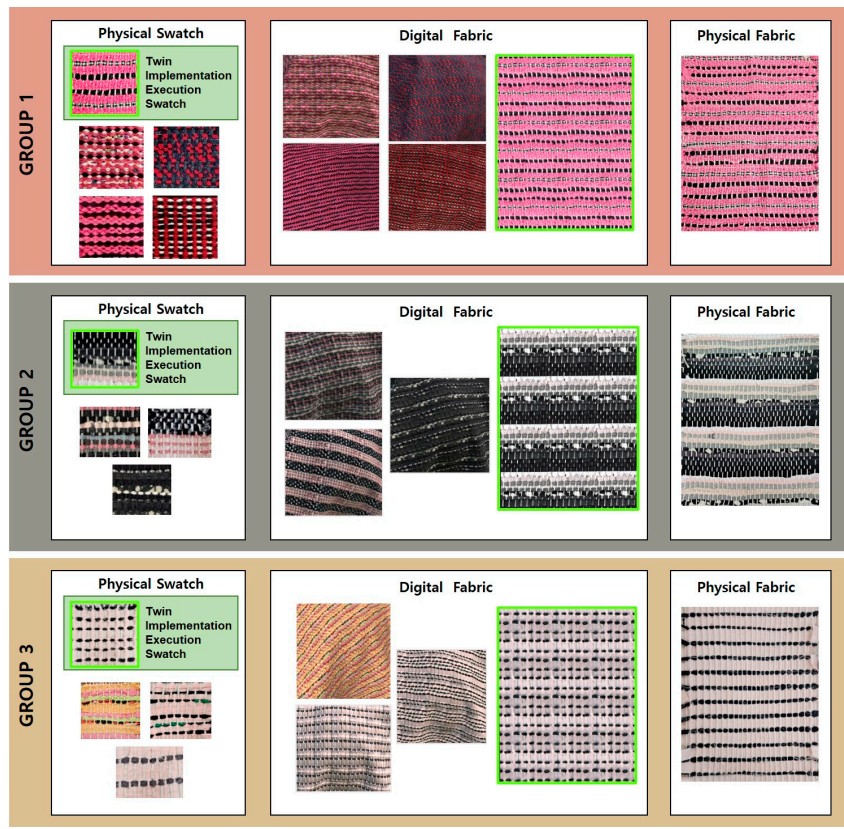

**Figure 9.** Results of the material development stage: developed physical swatches, digital fabrics, and physical fabrics.

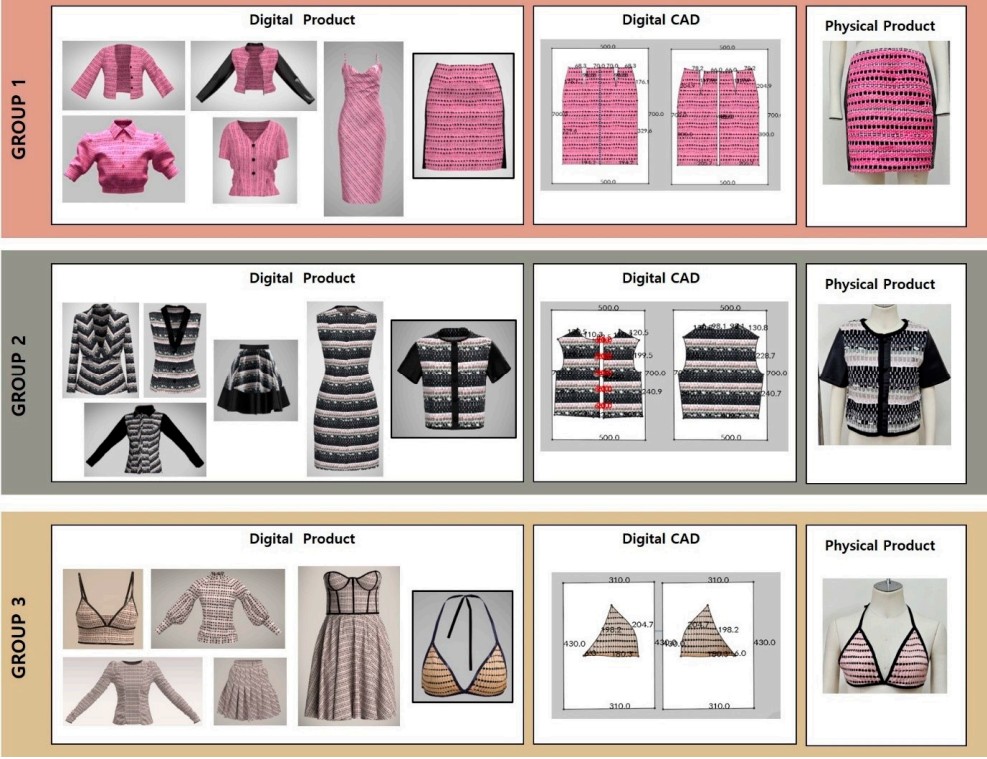

**Figure 10.** Results of the product manufacturing stage: digital products, capturing digital computer-aided design patterns, and physical products.

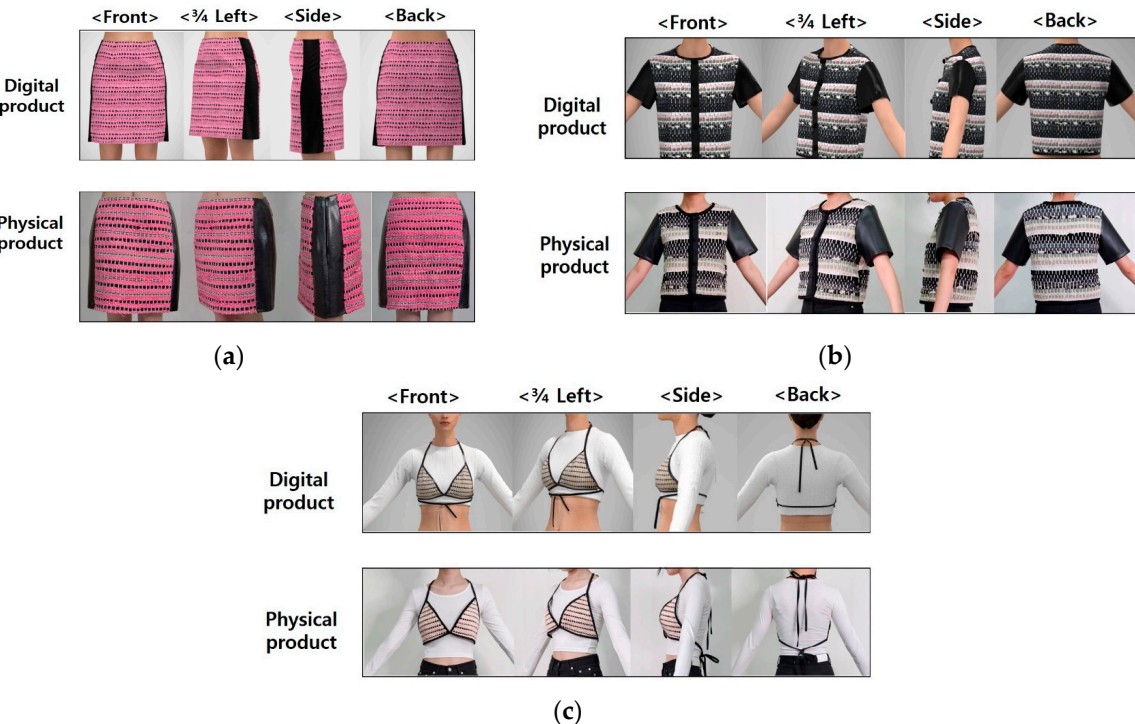

**Figure 11.** Comparison of photos of the human and avatar wearing the manufactured products: (**a**) skirt product in group 1, (**b**) jacket product in group 2, and (**c**) bustier bra product in group 3.

## 5. Evaluation and Discussion

Objective design evaluations of three pairs of physical and digital twin products were conducted, as well as in-depth expert evaluations through focus group interviews regarding each stage and the entire process. The results of the objective design evaluation are presented in Figure 12. The design characteristics of the physical and digital products were analyzed, and they had an average match rate of 90.1%. In particular, the material weave patterns, structural shapes, and garment silhouettes of the physical product and digital twins exhibited a high match of over 90%. Material colors, textures, and product details were less than 90% consistent, with judges explaining that this was due to differences in physical and digital lighting, reflectivity of materials, digital rendering, and differences in posture between a real person and a virtual avatar.

The following are the results of the in-depth evaluation based on the focus group interviews with experts. First, the evaluation results for each stage: In the first stage of material analysis, experts determined that clothing waste materials were suitable for use in products because of their stable supply, without the need to buy new ones, and because of the variety of materials. In other words, although the voluntarily collected discarded clothing could not play a role in the "product" due to the decline in fashionability, wearability, and functionality, it was a positive process for the design and material use by switching to the "material" perspective. The colorful patterns and designs, varying thicknesses, and details of clothing waste make it a unique woven material. Before moving to the design and material stages, clothing waste was evaluated for its suitability as a new material based on its size. Even if the size of the material was small, it could be mixed with other materials.

Second, in the design stage, both physical and digital designs were suitable based on the previously analyzed clothing waste data. The selection of waste materials for physical product fabrication based on material analysis data, prediction of the design by reflecting the color and texture of the waste materials, and planning of the product by dividing it into physical and digital methods were positively evaluated as circular and future-oriented methods that considered the use of the product after production.

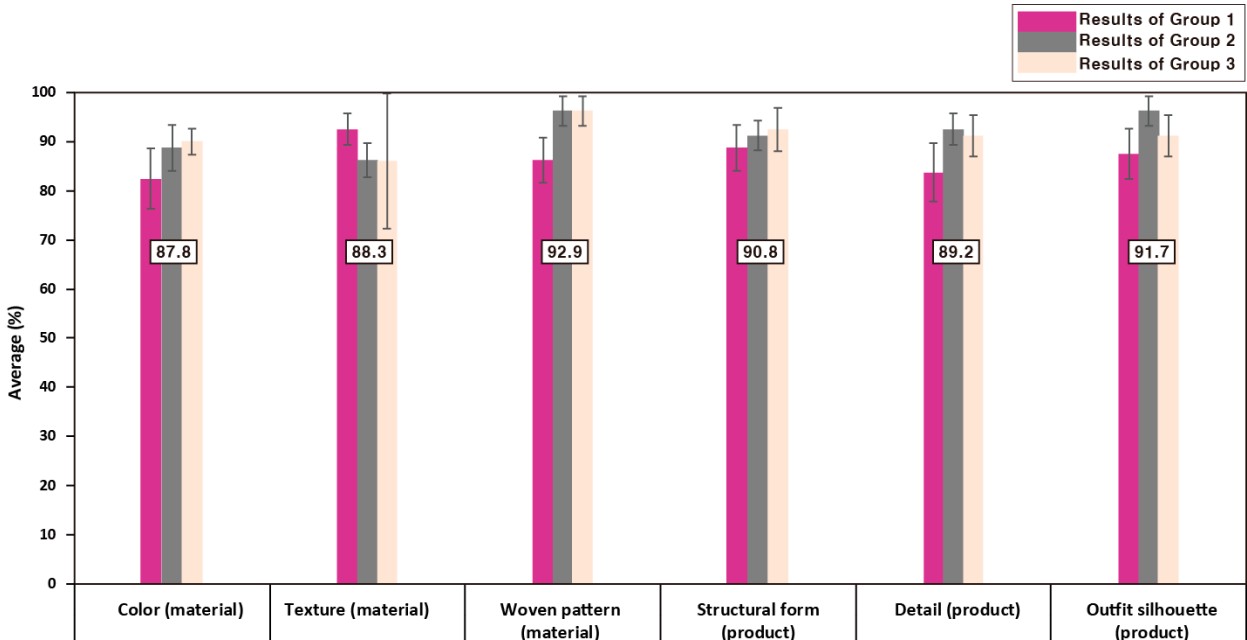

**Figure 12.** Results of the objective design evaluation: averaged evaluation scores for the color, texture, and woven pattern of the material, structural form and details of the product, and outfit silhouette.

In the material development stage, the process of preparing waste into woven material, which is a straightforward cutting and tape-production process, was evaluated as simple and easy. The weaving method is based on plain weaving, which is highly feasible for the general public. In a previous study [40], the process of materializing waste was somewhat difficult for the public to practice. However, the materialization method used in this study can significantly reduce the amount of waste, and was deemed a practical method that can educate consumers who generate waste. Additionally, the fabrics developed at this stage had a unique texture and weave, which translated the tweed from waste into a high-quality fabric.

At the product-manufacturing stage, the product was successfully completed while retaining the characteristics of the woven fabric and completely transforming the appearance of the waste material. The physical and digital products exhibited a high percentage of matching, which improved the accuracy of the pattern on the physical product that was successfully worn by virtual and real wearers.

The results of the in-depth evaluation of the entire process are shown in Table 5, and the specific explanations of the evaluation are as follows: First, regarding the appropriateness of the dual strategy, seven out of ten experts positively rated it as appropriate, while three experts rated it as inadequate and in need of further continuous improvement. The general opinion of the experts who answered "appropriate" was that the process sought to address clothing waste generated physically using digital methods in parallel with physical methods, thus unifying the goals of the physical and digital strategies. In addition, while 3D virtual clothing production in the fashion industry has thus far been limited to product prototyping, this process allows dual strategies to run parallel in all parts of the process, from material preparation to design, product development, and wearing, thereby creating a balanced mutual strategic relationship. In contrast, the experts who rated the product as "inappropriate" stated that although the digital design variations were many, limitations in design variation and physical production still existed. They suggested that more physical products made from existing clothing waste should be created and that experimental and feasible production methods should be further developed. In addition, experts have stated that 3D virtual digital product production technology cannot be conducted by the general public and should be accomplished in cooperation with specialized industries; therefore, the public practice method for production should also be considered.

**Table 5.** Results of the in-depth evaluation of process.

|  |  | The Answers from Experts (*n* = 10) | |
|---|---|---|---|
|  |  | Appropriate | Inappropriate |
| Appropriateness of the dual strategy | | 7 | 3 |
| Utilization of the process | Educational aspects | 8 | 2 |
|  | Industrial aspects | 5 | 5 |

The evaluation of the utilization of the project was divided into educational and industrial aspects. The results showed that eight out of ten experts rated the educational value of the program positively. The project was recognized for its potential to educate consumers about the value of clothing waste. It was found that consumers can directly observe the quality improvement of fabrics and products made from waste materials and that the easy weaving process will help provide non-majors interested in fashion with an easy methodology for creating upcycled products. The evaluation of the utilization of the project from the industrial side was 5:5, between positive and negative. On the positive side, the limitations of fabric size and product type were overcome to some extent by the digital strategy, and digital fabrics of different sizes could be applied to various types of products. By purchasing and wearing digital products made from physical clothing waste in a virtual digital environment, producers and consumers can cooperate to maintain a cleaner global environment, reduce clothing waste, extend PLCs, and create a virtuous cycle. However, for companies dealing with large amounts of clothing waste and inventory, the process of preparing waste into woven material must be conducted manually, which can result in additional labor costs. In addition, the limited size of the fabric restricts the types of products that can be manufactured, thus requiring additional technical considerations for competitive merchandising.

## 6. Conclusions

This study aimed to solve the problem of clothing waste disposal and material development to create a cleaner and more sustainable environment. A dual strategy that integrated both physical and digital approaches was adopted to achieve the objectives of clothing waste disposal and product development.

To improve the previous fashion product LCA, where the proportion of waste sent to landfills and incineration was higher than that recycled, the project conducted in this study connected the analysis of clothing waste, design, material development, and product manufacturing based on a method in which a large amount of waste can be used as product materials, as devised in previous studies [40,41]. Clothing waste was creatively used in large quantities as a material for physical and digital products, which solved the qualitative and quantitative limitations of these materials and products by digitally expanding them. Digital twin products shared the purpose of using products for resource circularity with physical products.

The limitations of this study are that the clothing waste collected during the material analysis stage was highly contaminated, and their material information and care labels were removed before it reached the author; the author acknowledges the study's limitations in accurately analyzing the chemical composition of the clothing waste materials. As a result, I propose leaving the precise chemical analysis of the available clothing waste materials and the assessment of the practicality and durability of the final physical product as future research topics.

This study explored the challenges of sustainable clothing waste disposal in the fashion industry. Although this study connected the disposal stage with the product manufacturing stage, it can still be extended to the merchandising and retail stages by field experts. This study could inspire the fashion industry to discover active alternatives for clothing waste disposal and accelerate the adoption of practical attitudes. As a material and design

professional, the author intends to create fabrics from different types of clothing waste that can be used to create a variety of fashion design products, conduct performance testing studies related to the durability, washability, and recontamination potential of the physical products created in this study, and explore educational programming strategies related to the project conducted in this study as well as modifications of the strategy to low- and middle-income countries.

**Funding:** This work was supported by the National Research Foundation of Korea (NRF), which is funded by the Korean government (MSIT) (Grant No.: RS-2022-00166075).

**Institutional Review Board Statement:** This study was approved and conducted under the supervision of the Catholic University's Ethics Committee (IRB) (IRB No. 040395-202207-01, approval date: 19 July 2022).

**Informed Consent Statement:** Informed consent was obtained from all subjects involved in this study.

**Data Availability Statement:** Not applicable.

**Acknowledgments:** I would like to thank my research students at the Catholic University of Korea, Yungkyu You, Dahyun Lee, and Seohyun Lee for their help in conducting this research.

**Conflicts of Interest:** The author declares no conflict of interest.

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
