# Peer review of "The Dual Strategy for Textile and Fashion Production Using Clothing Waste"

_sustainability, doi:10.3390/su151511509_

Round 1
Reviewer 1 Report
The topic of article is very good and scope of study is novel. However, there are following observations/suggestions.
· Can the author explain the benefits of dual strategy in terms of specific time and money i.e., how much time and money is saved by adopting this strategy
· The author has mentioned composition of fabrics but it would be better if can be specific about the composition i.e., how much percentage of cotton, polyester, rayon, silk, polyurethane acrylic etc in T shirts etc
· The author did not mention about the durability of clothes specially in terms of physical products. How much washing they can withstand.
· Figure 5 must mention the program used for creating digital textile.
· The author is also advised to mention the limitation of the current research work in the conclusion section .
· Author should double check the grammar, language editing needed.
Author should double check the grammar, language editing needed.
Author Response
First of all, I would like to thank you for your detailed reading and review of my manuscript. The manuscript has been organized according to your review. The following are my comments (blue letters in the report) and correction results for your review. The amendments to each comment are detailed below.
<Comments and Suggestions for Authors>
1) Can the author explain the benefits of dual strategy in terms of specific time and money i.e., how much time and money is saved by adopting this strategy
⇒ The 'Design & Production' process carried out in this study is not the entire Product Life Cycle (PLC) related to clothing waste management; rather, it is a part of it. As a result, it is practically challenging to measure specific cost savings and time reduction benefits derived from the dual-strategy approach. Numerous variables come into play, such as the type of product and production plans. However, if the process of this study were to be further expanded and applied to the entire fashion stream process, specific and minimum costs and time savings could be achievable within that scope.
In light of the reviewer's feedback, the devised dual strategy in this study, compared to the conventional fashion product Life Cycle Assessment (LCA), skips the 'raw material purchase' phase and instead utilizes clothing waste. As a result, there is a clear recognition of the potential benefits in terms of saving time and costs that would have been incurred in the conventional stages. In response to this feedback, the author has inserted the following content into the main text:
Line 203-207: (Inserted part) The main advantage of this process is that, unlike traditional LCA, no raw material purchases are made, thus saving time on material searching and expenses. The purpose of this process is to create new materials from clothing waste without the material purchase stage, and use them in products.
Furthermore, based on this study, it is deemed feasible for future research to explore an expanded process that includes marketing and retailing stages. In light of this possibility, the following content has been included in the conclusion.
Line 532-536: (Inserted part) This study explored the challenges of sustainable clothing waste disposal in the fashion industry. Although this study connected the disposal stage with the product manufacturing stage, it can still be extended to the merchandising and retail stages by field experts. This study could inspire the fashion industry to discover active alternatives for clothing waste disposal and accelerate the adoption of practical attitudes.
2) The author has mentioned composition of fabrics but it would be better if can be specific about the composition i.e., how much percentage of cotton, polyester, rayon, silk, polyurethane acrylic etc in T shirts etc
⇒ During the Material Analysis Stage, I attempted to examine the attached tags and care labels on clothing waste pieces and organize the information gathered. However, I encountered practical limitations in obtaining accurate analyses. Over two-thirds of the collected clothing waste pieces had faded or missing text on their labels, likely due to the nature of waste disposal and severe contamination, making it challenging to make accurate assessments. As a result, in Table 3, under the "T-shirts" section for fabric composition, I indicated "no info" to represent cases where the label information was absent. The information presented in the table only includes cases where fabric materials were clear either from the attached labels or through visual inspection. While there were some instances where material proportions were specified on the labels, these occurrences were scarce in the collected clothing waste samples. Therefore, it was not possible to generalize the fabric composition information across all items used in this study. Consequently, this specific information was not included in the paper.
While considering the reviewer's feedback, I acknowledged that readers would likely be curious about the limitations regarding the material analysis in this study. Hence, in light of this, I have added content in the conclusion section to address the constraints related to the material information analysis in this paper.
Line 527-531: (Inserted part) The limitations of this study are that the clothing waste collected during the material analysis stage was highly contaminated, and their material information and care labels were removed before it reached the authors; therefore, performing an accurate fiber ratio analysis was not possible.
3) The author did not mention about the durability of clothes specially in terms of physical products. How much washing they can withstand.
⇒ The physical strength, including durability testing of the clothing, was not included in the current study. This research focuses on utilizing waste materials to create fabrics, designing products, and taking them through the process of actual product development. However, I am planning future research to conduct various physical characteristic tests, including the durability of the garments created through this study. Accepting the reviewer's feedback, I have now provided further clarification in the conclusion section regarding the limitations and future research direction:
Line 527-542: (especially yellow highlight part) The limitations of this study are that the clothing waste collected during the material analysis stage was highly contaminated, and their material information and care labels were removed before it reached the authors; therefore, performing an accurate fiber ratio analysis was not possible. Further durability evaluation is needed to ensure the practicality and stable use of the final physical product.
This study explored the challenges of sustainable clothing waste disposal in the fashion industry. Although this study connected the disposal stage with the product manufacturing stage, it can still be extended to the merchandising and retail stages by field experts. This study could inspire the fashion industry to discover active alternatives for clothing waste disposal and accelerate the adoption of practical attitudes. As a material and design professional, the author intends to create fabrics from different types of clothing waste that can be used to create a variety of fashion design products, conduct performance testing studies related to the durability, washability, and recontamination potential of the physical products created in this study, and explore educational programming strategies related to the project conducted in this study as well as modifications of the strategy to low- and middle-income countries.
4) Figure 5 must mention the program used for creating digital textile.
⇒ As you suggested, I added the program name to the caption of Figure 5.
Line 323: Figure 5. Screenshot of the Adobe Substance 3D Sampler program used to create digital textiles.
5) The author is also advised to mention the limitation of the current research work in the conclusion section.
⇒ Upon reevaluating my research based on your previous questions, I have become aware of the limitations in the current study. Specifically, the challenges related to material analysis, particularly in analyzing fiber blends due to textile contamination and removal difficulties, have been acknowledged as a significant constraint. Additionally, the absence of durability testing for the products poses a need for additional processes to ensure greater reliability in product utilization. I wrote about this in the conclusion part.
Line 527-531: (Inserted part) The limitations of this study are that the clothing waste collected during the material analysis stage was highly contaminated, and the material information and care labels on the clothing waste were removed before it reached the authors, so it was not possible to perform an accurate fiber ratio analysis. Furthermore, further evaluation of durability is needed to ensure the practicality and stable use of the final physical product.
6) Author should double check the grammar, language editing needed.
⇒ This manuscript received English proofreading services before submission. However, I followed the reviewer's advice and had it language proofread once more before submitting the revision report. A screen capture of the certificate has been inserted into the report.

Reviewer 2 Report
An interesting read on how digital and physical approach can be used in minimising clothing waste. The authors have published two studies prior to this research. Methdology undertaken was thorough and the results are well explained with detail comparison between the two approaches. Please see comments below for amendments to improve quality of this paper:
· In abstract, for the last part – please conclude how does this study helps in term of sustainability aspects of fashion industry.
· In the introduction, please explain how the digital prototype / design phase can help in the circularity i.e. solving waste issue
· There are errors in numbering in Line 59 onwards
· Figure 2a and Figure 2b – please check spelling
· In Line 306, please specify types of clothing waste asked from the subjects. Random or any specific types?
· Table 2 – the country should be United States
· It would be best if the results from Line 521 to Line 555 can be presented in a form of table of chart.
· The authors should explain further on how the twin strategy adopted can be used to materialize the circular economy agenda.
English is good, just do proofread for some spelling errors. Spelling rrrors spotted in some figures.
Author Response
First of all, I would like to thank you for your detailed reading and review of my manuscript. The manuscript has been organized according to your review. The following are my comments (blue letters in the report) and correction results for your review. The amendments to each comment are detailed below.
<Comments and Suggestions for Authors>
(The '⇒' after this arrow is the author's answer)
1) In abstract, for the last part – please conclude how does this study helps in term of sustainability aspects of fashion industry.
⇒I have added how this research contributes to the sustainability of the fashion industry in the final part of the abstract
Line 19-21: (Inserted part) Therefore, this study connected the quantitative disposal of garment waste to the qualitative design and production of new material, introducing a new process strategy to maintain sustainability in the fashion industry.
2) In the introduction, please explain how the digital prototype / design phase can help in the circularity i.e. solving waste issue
⇒ Until now, the digital prototyping and design processes in the fashion industry stream primarily focused on designers' free-spirited ideas, aiming for challenging and experimental design prototypes, with little consideration for the final product's disposal. In contrast, this researcher proposes a dual strategy attempting to connect the digital prototyping and design processes with the ultimate waste disposal stage, both physically and digitally. By doing so, the quantitative challenges in waste management can be fundamentally addressed, while achieving a forward-looking 'sustainability' that is not only linked to the qualitative aspects of design but also to the sustainable future. The content regarding this proposal has been supplemented in the introduction part of the research
Line 84-105: (especially yellow highlight part)
The beginning of fashion production is often a digital prototype, and the end is a physical product that has been used and lost value. To connect the end and beginning stages flexibly, the product strategy for each stage deliverable should be interconnected and complementary. To date, the digital prototyping and design stage has been the furthest from the disposal of physical products, with challenging and experimental prototyping based on free-form ideas using infinitely new raw materials in a digital environment. The technical advantages of a digital strategy do not translate into the later stages of physical production and inventory handling, and even less so for product disposal. As stated in the EU report [32], the need to develop strategies to transition the end of the fashion stream from simple incineration and landfilling to fiber-to-fiber recycling and digital textiles from physical waste is urgent. The final stage of the product, that is, the quantitative treatment of physically generated waste, should be linked to the first stage of the product, the qualitative design process of product development. In other words, the direction of materializing waste and using it as a raw material for products should be explored, and a twin strategy should be implemented to increase the viability of the strategy. In the future, the fashion industry will need to implement the first twin strategy, which connects the garment disposal phase with the garment design phase, and the second twin strategy, which involves the simultaneous implementation of physical and digital methods to connect the two phases. Achieving the twin strategy is not only an alternative to the traditional disposal of clothing waste in landfills and incineration, but also a way to complete the sustainable cycle of the fashion industry, which has been accelerated by digitalization.
3) There are errors in numbering in Line 59 onwards
⇒ The numbering of references has been double-checked and re-numbered.
4) Figure 2a and Figure 2b – please check spelling
⇒ I have corrected the typo in Figure 2. The corrected part is “Distribution”.
5) In Line 306, please specify types of clothing waste asked from the subjects. Random or any specific types?
⇒ The clothing waste was collected randomly without any specific restrictions. I have further reinforced this information as follows
Line 267-268: (Inserted part) Clothing waste, with no restrictions on type, were voluntarily collected from subjects one week prior to the project.
6) Table 2 – the country should be United States
⇒ Out of the ten experts who participated in the evaluation of this research, two individuals held American nationality, which is specified in Table 1. The non-American evaluators also demonstrated active engagement in their respective countries and internationally, and all of them possessed at least ten years of profound expertise in their respective fields. Supplementary information regarding these details has been inserted into the main text.
Line 337-339: (Inserted part) The experts have at least 10 years of specialized knowledge and experience in fashion design, materials, and products, and were active in both domestic and international research and industry
7) It would be best if the results from Line 521 to Line 555 can be presented in a form of table of chart.
⇒ As advised by the reviewer, Table 5 was added. (Line 474)
8) The authors should explain further on how the twin strategy adopted can be used to materialize the circular economy agenda.
⇒ I hope that this research can actively explore viable alternatives for clothing waste management across the entire fashion industry and accelerate the adoption of practical approaches. As a contribution from experts in merchandising and retailing, I aspire to see this process expanded and utilized. This has been added to the conclusion.
Line 532-542: (Inserted part) This study explored the challenges of sustainable clothing waste disposal in the fashion industry. Although this study connected the disposal stage with the product manufacturing stage, it can still be extended to the merchandising and retail stages by field experts. This study could inspire the fashion industry to discover active alternatives for clothing waste disposal and accelerate the adoption of practical attitudes. As a material and design professional, the author intends to create fabrics from different types of clothing waste that can be used to create a variety of fashion design products, conduct performance testing studies related to the durability, washability, and recontamination potential of the physical products created in this study, and explore educational programming strategies related to the project conducted in this study as well as modifications of the strategy to low- and middle-income countries.
9) English is good, just do proofread for some spelling errors. Spelling errors spotted in some figures.
⇒ This manuscript received English proofreading services before submission. However, I followed the reviewer's advice and had it language proofread once more before submitting the revision report. A screen capture of the certificate has been inserted into the report.

Reviewer 3 Report
In the manuscript entitled "A Study on the Dual Strategy for Textile and Fashion Production Using Clothing Waste" the authors have focused on two important aspects physical strategy and digital strategy. The manuscript is well written, well structured and well researched. I recommeded the work for publications after minor modifications. The comments are appended below
(1) Although the authors have discussed the physical strategy and digital strategy for recycling of clothing waste. However in real sceanrio the digital strategy is seldom found/not found especially in the second and third world countries. So authors should discuss this aspect as a challenge in the conclusion part.
(2) In Figure 12 the left Y axis missing the tick mark. Alo the quality of the figure 12 should be improved. In figure 12 looks like all error bars are same, is it true?? Statistical significance should be determined for the graphs.
(3) A graphical abstract should be prepared.
Author Response
First of all, I would like to thank you for your detailed reading and review of my manuscript. The manuscript has been organized according to your review. The following are my comments (blue letters in the report) and correction results for your review. The amendments to each comment are detailed below.
<Comments and Suggestions for Authors>
The '⇒' after this arrow is the author's answer
(1) Although the authors have discussed the physical strategy and digital strategy for recycling of clothing waste. However in real sceanrio the digital strategy is seldom found/not found especially in the second and third world countries. So authors should discuss this aspect as a challenge in the conclusion part.
⇒ As per the reviewer's suggestion, the dual strategy for the second and third worlds presents a new challenge, as well as a promising topic for future research that requires appropriate modifications and improvements to the current process. In light of the reviewer's feedback, I have incorporated the convergence of their opinions into the conclusion, outlining my future research plans.
Line 536-542: (Inserted part) As a material and design professional, the author intends to create fabrics from different types of clothing waste that can be used to create a variety of fashion design products, conduct performance testing studies related to the durability, washability, and recontamination potential of the physical products created in this study, and explore educational programming strategies related to the project conducted in this study as well as modifications of the strategy to low- and middle-income countries.
(2) In Figure 12 the left Y axis missing the tick mark. Alo the quality of the figure 12 should be improved. In figure 12 looks like all error bars are same, is it true?? Statistical significance should be determined for the graphs.
⇒ Thank you so much for your detailed advice. I found an incorrect setting of the error bars in my table. The graph was redrawn and the quality improved. (Line 440-443)
(3) A graphical abstract should be prepared.
⇒ I prepared a graphic abstract as advised. My graphical abstract is Figure 1, which I will upload with the manuscript.

Round 2
Reviewer 1 Report
Authors mentioned that could not find tags to check material composition, my suggestion for future is to perform chemical analysis to test fabric composition in such cases. overall, revised version is fine and can be accepted.
Author Response
Dear Sustainability Journal Reviewer.
I would like to thank you for the opportunity to submit this final, minor revision of this manuscript.
I have once again addressed the reviewer' points in the conclusion sectopm and revised the manuscript.
The revisions to the reviewers' comments and my comments are listed below.
=================================================
<Comments and Suggestions for Author>Authors mentioned that could not find tags to check material composition, my suggestion for future is to perform chemical analysis to test fabric composition in such cases. overall, revised version is fine and can be accepted
⇒ Author's comment:
As the reviewer suggests, I agree that the material needs to be analyzed chemically and precisely, and I have reworded this as a limitation.
I revised and expanded the study limitations section of the "Conclusion" as follows (especially the green text).
Line 528-532:
The limitations of this study are that the clothing waste collected during the material analysis stage was highly contaminated, and their material information and care labels were removed before it reached the author; The author acknowledges the study's limitations in accurately analyzing the chemical composition of the clothing waste materials. As a result, I propose leaving the precise chemical analysis of the available clothing waste materials and the assessment of the practicality and durability of the final physical product as future research topics.